# Coordination environment dependent selectivity of single-site-Cu enriched crystalline porous catalysts in $CO_2$ reduction to $CH_4$

Yu Zhang[1,3], Long-Zhang Dong[1,3], Shan Li[1], Xin Huang[1], Jia-Nan Chang[1], Jian-Hui Wang[1], Jie Zhou[2], Shun-Li Li[1] & Ya-Qian Lan [1,2 ✉]

The electrochemical $CO_2$ reduction to high-value-added chemicals is one of the most promising and challenging research in the energy conversion field. An efficient ECR catalyst based on a Cu-based conductive metal-organic framework (Cu-DBC) is dedicated to producing $CH_4$ with superior activity and selectivity, showing a Faradaic efficiency of $CH_4$ as high as ~80% and a large current density of $-203 \, mA \, cm^{-2}$ at $-0.9 \, V$ vs. RHE. The further investigation based on theoretical calculations and experimental results indicates the Cu-DBC with oxygen-coordinated Cu sites exhibits higher selectivity and activity over the other two crystalline ECR catalysts with nitrogen-coordinated Cu sites due to the lower energy barriers of $Cu-O_4$ sites during ECR process. This work unravels the strong dependence of ECR selectivity on the Cu site coordination environment in crystalline porous catalysts, and provides a platform for constructing highly selective ECR catalysts.

[1] Jiangsu Collaborative Innovation Centre of Biomedical Functional Materials, Jiangsu Key Laboratory of New Power Batteries, School of Chemistry and Materials Science, Nanjing Normal University, Nanjing 210023, P. R. China. [2] School of Chemistry, South China Normal University, Guangzhou 510006, P. R. China. [3] These authors contributed equally: Yu Zhang, Long-Zhang Dong. ✉email: yqlan@njnu.edu.cn

The effectively electrocatalytic conversion of the main greenhouse gas, carbon dioxide ($CO_2$), is of great significance to reduce the negative impact of climate change as well as to mitigate the energy crisis caused by the shortage of fossil fuels[1–3]. In recent years, an increasing number of research works have been devoted to acquiring well-performed $CO_2$ reduction electrocatalysts to convert $CO_2$ into high-value-added chemicals, such as carbon monoxide, formate, acetate, hydrocarbons, alcohols, etc.[4,5]. Among the electrochemical $CO_2$ reduction (ECR) products, hydrocarbons, especially $C_2H_4$ and $CH_4$, have attracted great attention due to their compatibility with existing infrastructure and the promising substitution for fossil fuels[6]. However, the sluggish kinetics and side reactions of ECR lead to low selectivity, plain activity, and limited durability. The activation of $CO_2$ is hindered by the extremely stable molecular structure with $C=O$ dissociation energy as high as 803 kJ mol$^{-1}$ (refs. [7,8]). In addition, resulting from the interference of competitive hydrogen evolution reaction (HER) and the multiple electron transfer processes involved during the reaction, the ECR-to-hydrocarbons suffers from low selectivity since different pathways occur under similar reduction potentials with various products[8–10]. To date, Cu-based electrocatalysts, including Cu oxides[11,12], metallic Cu[13,14], and Cu-containing molecules[15], have been proved to be capable of promoting ECR toward hydrocarbons and alcohol products at acceptable FEs. However, ECR selectivity towards hydrocarbons is still hindered by the lack of intelligent electrocatalysts.

With this aspect, the design of single-site catalysts (SSCs) is a promising strategy to achieve precise catalysis with high selectivity due to their well-defined, isolated and atomically dispersed single-metal active sites[16–19]. As typical SSCs, the single-atom catalysts have been widely investigated in the ECR field[20–24]. However, most of the reported single-atom catalysts tend to form two-electron-product CO rather than hydrocarbons due to the favorable energy barrier of CO production as well as the highly dispersed and low loaded catalytic sites[25]. In addition, most of the reported single-atom sites are nitrogen-coordinated, usually M-$N_xC_{4-X}$, and the other heteroatom coordinated single-metal sites are seldom studied[22,26]. And it probably occurs the uncontrollable coordination environment around the metal sites and the uneven dispersion of catalytic sites within the catalysts as single-atom catalysts are usually obtained from the carbonization of the metal-containing precursors[26]. Another major category of SSC is the crystalline materials with well-defined structures and uniformly arranged single-metal sites, usually metal-organic framework (MOF), covalent organic framework (COF), or metal-organic complex materials[18,27,28]. The structure and metallic site coordination are controllable while the low electrical conductivity is still a tough challenge to the crystalline ECR catalysts.

Conductive metal-organic framework (cMOF) materials, a subclass of MOF constructed by self-assembling transition metal ions with conjugated organic ligands, are promising SSC candidates for selective ECR studies due to the unique redox and electrical conductivity as well as their MOF-based features including the unsaturated-coordinated metal sites and versatile porosity[29,30]. Additionally, the well-defined structure and excellent tunability of cMOF are also adequate for establishing an accurate structure–activity relationship for precise catalysis. A series of metallophthalocyanine (MPc)-based conductive MOFs was designed to dominate the activity and selectivity of ECR-to-CO by adjusting the metal site within MPc and heteroatomic linkage[31]. Feng et al. also reported a bimetallic two-dimensional cMOF embedded with phthalocyanine Cu–$N_4$ sites and zinc-bis(dihydroxy) Zn–$O_4$ sites for the selectivity modulation towards ECR-to-CO and HER[32]. More recently, a nitrogen-rich cMOF with graphene-like porous structure, $Cu_3(HHTQ)_2$, was

synthesized and served as ECR catalysts, showing a high selectivity to $CH_3OH$ with Faradaic efficiency (FE) reached up to 53.6%[33]. Despite the progress in ECR applications with cMOF, there is still few research on cMOF electrocatalysts for efficient ECR-to-hydrocarbons with enhanced activity and satisfactory hydrocarbons selectivity.

Herein, a Cu-based cMOF composed of the highly conjugated graphene-like ligand (dibenzo-[g,p]chrysene-2,3,6,7,10,11,14,15-octaol, 8OH-DBC) and Cu nodes is served as efficient electrocatalyst for $CO_2$ reduction. The highly conjugated organic ligand endows the Cu-DBC with unique redox property and electrical conductivity. The abundant and uniformly distributed Cu–$O_4$ sites are beneficial to the effective ECR-to-$CH_4$ process with high selectivity. It exhibits a high $CH_4$ FE of ~80% at a low reduction potential of −0.9 V vs. RHE with a partial current density of −162.4 mA cm$^{-2}$, which is among the best Cu-based electrocatalysts for $CO_2$ reduction to $CH_4$. In addition, the relationship between the coordination environment of single Cu site and the selectivity of electroreduction catalysis is investigated based on the well-defined structure of the crystalline porous catalysts. The particularity of Cu-DBC catalyst towards ECR-to-$CH_4$ reaction and the detailed catalytic mechanism are further systematically analyzed via electrocatalytic measurements and computational studies. This research provides a platform for the design of ECR catalyst with well-defined structure and establishes foundation to accurate structure–reactivity correlations for the construction efficient ECR catalysis.

## Results

**Synthesis and characterizations of Cu-DBC electrocatalyst**. The Cu-DBC electrocatalyst was synthesized via a solvothermal method by the conjugated ligand of 8OH-DBC and copper acetate at 85 °C in the mixed solvent of N,N-dimethylformamide and water[34]. The structure details of the resulting Cu-DBC sample are shown in Fig. 1a and Supplementary Fig. 1. Each Cu ion coordinates with two catechol groups from two different 8OH-DBC ligands to form a Cu–$O_4$ site, and each 8OH-DBC ligand connects with four Cu ions as a 4-c node. By these connection modes, the three-dimensional frameworks and one-dimensional channels with pore size of ~1.0 nm are formed and shown in Fig. 1a. Additionally, the topological analysis indicates that Cu-DBC presents a fourfold interpenetration dia topology (Supplementary Fig. 1)[34]. For the gas adsorption test, the solvent exchange has been taken previously by soaking Cu-DBC powder in pure methanol with several repeats during 3 days. The powder X-ray diffraction (PXRD) pattern displayed in Fig. 1b shows that the synthesized sample is in good agreement with the simulated pattern, confirming the successful synthesis of Cu-DBC and well crystallinity. The nitrogen adsorption–desorption isotherms shown in Supplementary Fig. 2 exhibit an isotherm with an obvious adsorption in $P/P_0$ range <0.1, revealing the presence of microporous and the BET surface area is 133 m$^2$ g$^{-1}$ with a total pore volume of 0.29 cm$^3$ g$^{-1}$. The pore size distribution displays the main pore width of 1.03 nm, which is consistent with the simulated structure model. $CO_2$ adsorption tests at 298 K revealed that the as-prepared Cu-DBC also exhibits $CO_2$ capture capability with an uptake capacity of 38.7 cm$^3$ g$^{-1}$ (Supplementary Fig. 3).

The charge delocalization between metal ions and conjugated ligands endows conductive MOF with electrical conductivity as well as unique redox activity, making the conductive MOF promising platform for electrocatalysis and energy storage applications[29,30,35]. The conductivity of the prepared Cu-DBC was measured by the four-contact probe method. As calculated from the current–voltage characteristic in Fig. 1c, Cu-DBC exhibits an electrical conductivity of $1.2 \times 10^{-2}$ S m$^{-1}$ which is

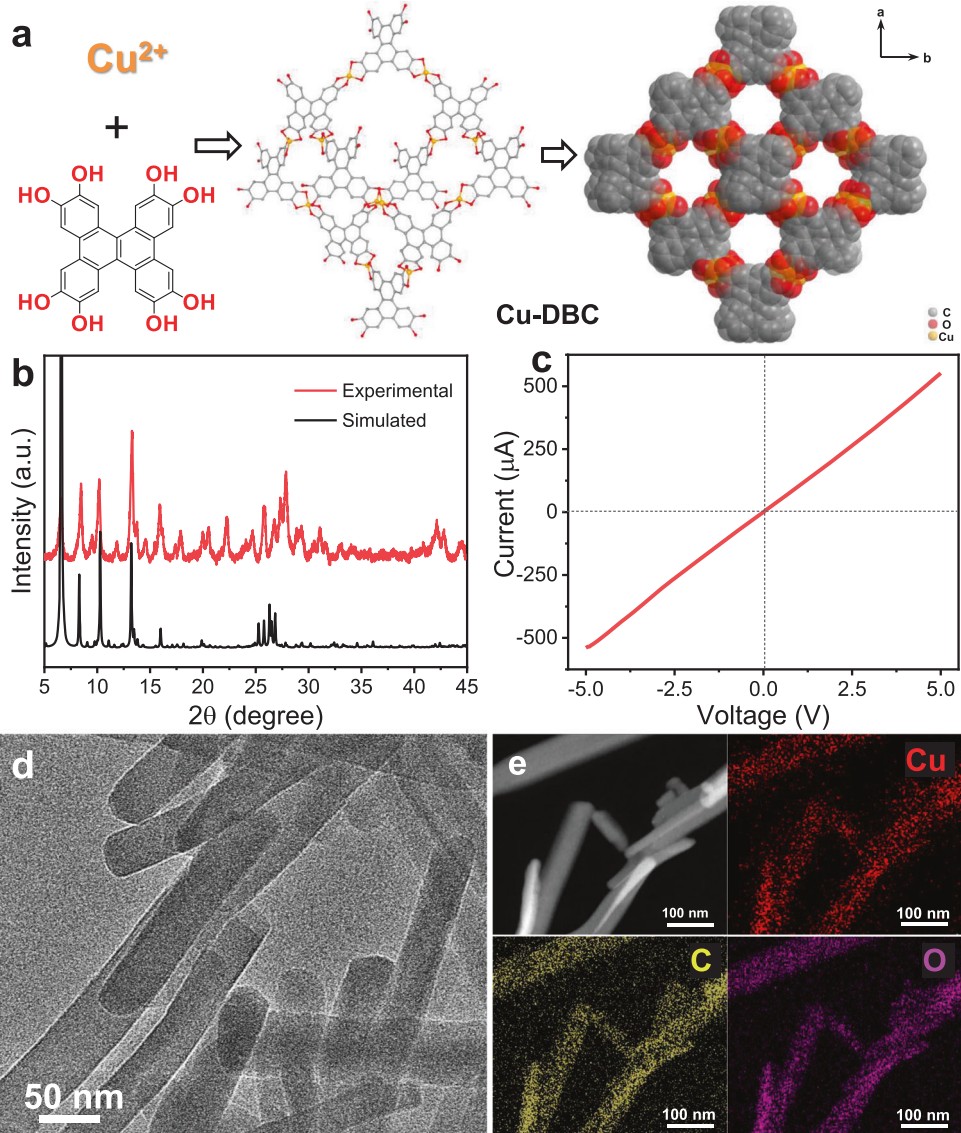

**Fig. 1 Characterizations of Cu-DBC electrocatalyst. a** The structure obtained by Cu ions and 8OH-DBC. **b** Simulated and experimental PXRD patterns. **c** Current–voltage characteristic using four-contact probe method. **d** TEM image. **e** STEM image and corresponding element mapping images of Cu, C, and O.

much higher than those of the conventional MOFs without conjugated ligands[34,36]. The scanning electron microscopy (SEM) and transmission electron microscopy (TEM) images shown in Supplementary Fig. 4 and Fig. 1d suggest that the resulting Cu-DBC displays a stick-like morphology with the length ranging from 500 nm to 1 μm and a width varying from 50 to 100 nm. The composition of the Cu-DBC is revealed by scanning transmission electron microscopy (STEM) with an energy-dispersive X-ray spectrum (EDS), which confirms the homogeneous distribution of Cu, C, and O in Cu-DBC nanorods (Fig. 1e and Supplementary Fig. 5). The state of Cu species is revealed by X-ray photoelectron spectroscopy (XPS), as shown in Supplementary Fig. 6. There are no Cu(0) species in Cu-DBC resulting from the Cu LMM Auger spectrum[12,37]. Cu 2p spectrum exhibits a dominant peak assigned to Cu(II) centered at ~933.77 eV and a trace peak for Cu(I) ~932.18 eV[12,34]. The small amount of Cu(I) species originates from the charge compensation of Cu(II) by the redox-active nature of the ligands to keep the whole structure in charge balance[34]. The TGA analysis was taken under Ar to show the thermal stability of Cu-

DBC (Supplementary Fig. 7). The specific content of Cu element in the as-synthesized Cu-DBC is 20.01 wt% as determined by ICP-AES (Supplementary Table 1). The uniformly arranged structure, conductive skeleton and enrich Cu sites of Cu-DBC suggest that it probably has the potential to serve as an electrocatalyst for $CO_2$ reduction.

**ECR-to-CH4 performance of Cu-DBC electrocatalyst**. To evaluate the ECR performance of the as-synthesized Cu-DBC, all electrochemical tests were carried out with a three-electrode flow cell setup using Pt foil as the counter electrode and Ag/AgCl electrode as the reference electrode. The anion exchange membrane is used to separate the cathode and anode. The catalysts were coated onto gas-diffusion layer (GDL)-modified carbon paper to act as work electrodes. Compared with the conventional H-type cell in which $CO_2$ is dissolved in weakly alkaline bicarbonates solution (usually $KHCO_3$ or $NaHCO_3$(aq)) and exists in the form of hydrated $HCO_3-$, the flow cell is gas-fed using alkaline solution with flowing $CO_2$ gas pass through the GDL[38–40]. The $CO_2$ mass transfer to catalyst surface is greatly

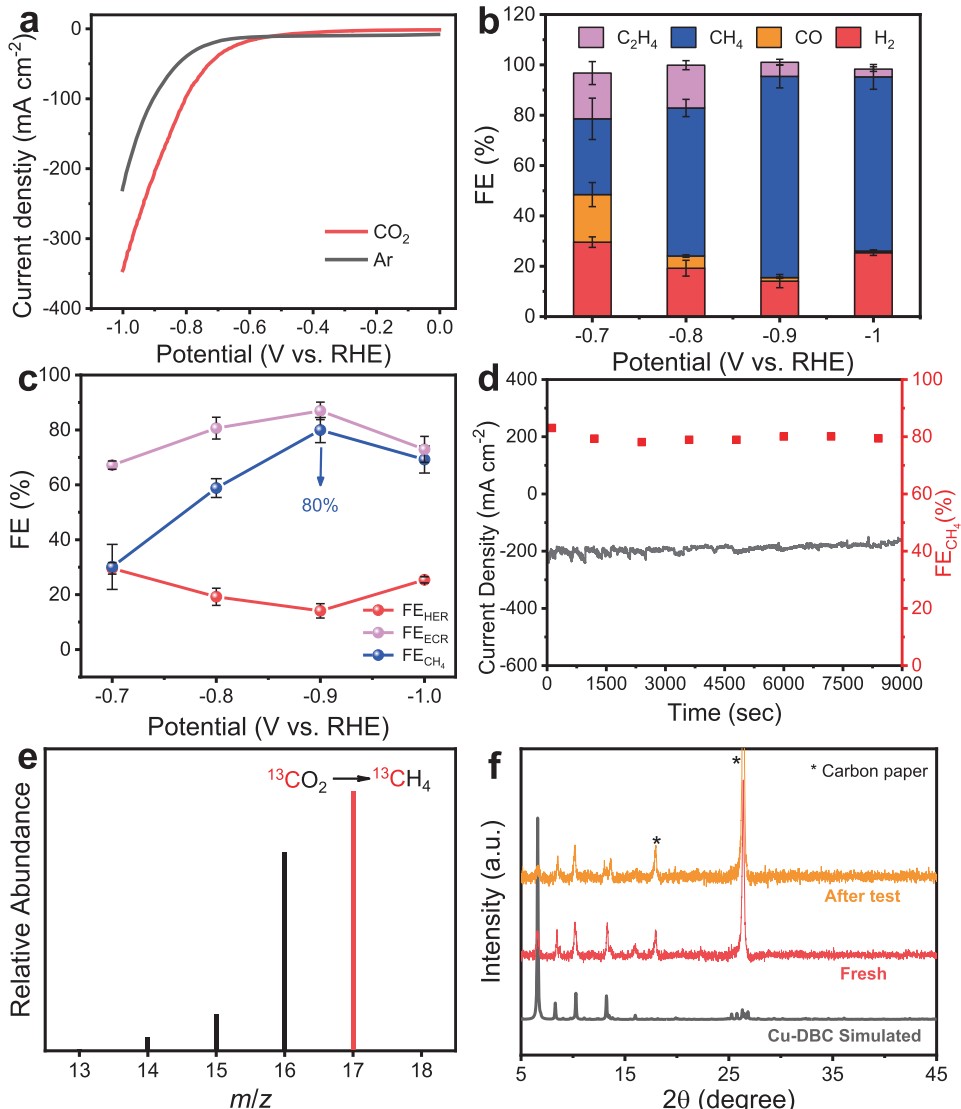

**Fig. 2 ECR-to-CH₄ performance of Cu-DBC electrocatalyst. a** Polarization curves recorded under CO₂ and Ar. **b** Faradaic efficiencies (FEs) of ECR products at different applied potentials. **c** FEs for HER, ECR, and CH₄ recorded at different applied potentials. **d** Current profile and FEs of CH₄ at a constant voltage of −0.9 V vs. RHE. **e** The mass spectra of ¹³CH₄ recorded under ¹³CO₂ atmosphere. **f** XRD patterns of fresh and tested Cu-DBC modified GDL-carbon paper electrodes.

enhanced with GDL, thus leading to substantially large current densities[39]. The detailed scheme of the GDL and flow cell test setup is presented in Supplementary Fig. 8.

Linear sweep voltammetry (LSV) polarization curves measured under flowing CO₂ and Ar atmosphere at a wide potential range from 0 to −1.0 V vs. RHE are shown in Fig. 2a. It demonstrates that Cu-DBC catalyst delivers lower onset potential and larger current densities in flowing CO₂ than those in Ar, indicating the possibility of higher activity of ECR than HER. ECR experiments were carried out at reductive potentials varying from −0.7 to −1.0 V vs. RHE. To evaluate the selectivity during ECR reaction, the gas products generated from electrocatalysis were detected by online gas chromatography (GC) system. The standard curves on GC were tested and shown in Supplementary Tables 2 and 3 and Supplementary Fig. 9. And the related gas chromatograms of flame ionization detector (FID) and thermal conductivity detector (TCD) are displayed in Supplementary Fig. 10. As shown in Fig. 2b, C₂H₄, CH₄, CO, and H₂ are detected to be products in the system. The FEs of products at different applied potentials are calculated and the sum of FE is found to be ~100% at all

potentials. The total FEs of ECR are larger than those of HER over the entire applied potential range and reach up to ~86.9% at −0.9 V vs. RHE, suggesting higher selectivity of Cu-DBC catalyst towards ECR than that of HER (Fig. 2c). In addition, CH₄ is the main ECR product among all. The FE for CH₄ production over Cu-DBC electrocatalyst is increased over the applied potential ranging from −0.7 to −0.9 V vs. RHE, and impressively, it raises to a maximum value of ~80.0% at −0.9 V vs. RHE (Fig. 2c), which is one of the best ECR-to-CH₄ performances to date. As shown in Supplementary Table 6, the Cu-DBC catalyst delivers larger active current density with a high CH₄ selectivity at a low reductive potential compared to the reported Cu-based ECR-to-CH₄ electrocatalysts tested in both flow cell setup[38,39] and H-type cell[41,42].

To further investigate the activity of Cu-DBC catalyst to ECR, chronoamperometric curves at various potentials on Cu-DBC were measured and shown in Supplementary Fig. 11. All the current densities obtained under potentials varying from −0.7 to −1.0 V can approximately keep unchanged, disclosing that the catalytic activity of Cu-DBC can maintain stable. Notably, it can

reach up to high current densities of approximately $-200$ and $-360 \, mA \, cm^{-2}$ at $-0.9$ and $-1.0 \, V$ vs. RHE. The long-term durability under the optimal potential of $-0.9 \, V$ vs. RHE was performed via chronoamperometric test to assess the stability of the Cu-DBC catalyst. As shown in Fig. 2d, Cu-DBC shows a negligible decrease in active current density during the 9000-s continuous electrolysis. Furthermore, the FE of $CH_4$ detected by the online GC system every 1200 s exhibits slight decay compared with the initial one and kept at ~80% during the stability test. The [1]H nuclear magnetic resonance spectroscopy indicates that there were hardly liquid products obtained from the electrocatalysis under the optimal test conditions (Supplementary Fig. 12). In addition, the isotopic-labeling experiment using $^{13}CO_2$ was executed under identical conditions as for the $^{12}CO_2$ experiment to verify the carbon source of the gas products evolved from $CO_2$ reduction. All products were analyzed by gas chromatography and mass spectrometry (GC-MS). As displayed in Fig. 2e and Supplementary Fig. 13, the signal peaks at $m/z = 17$, 29, and 30 are attributed to $^{13}CH_4$, $^{13}CO$, and $^{13}C_2H_4$, respectively, confirming that the carbon-containing gas products are all converted from $CO_2$.

Although many MOF- or COF-based ECR catalysts with unchanged structure after ECR tests have been reported[31–33], the phase transition might occur in some crystalline materials during electrocatalysis, leading to the change of the instinct catalytic sites[28,38,42]. Thus, the structural stabilities of the Cu-DBC electrocatalyst before and after electrocatalysis have been characterized and investigated. The XRD patterns shown in Fig. 2f suggest that there is no phase transition or obvious structural change after electrocatalytic tests. Furthermore, the morphologies of Cu-DBC still maintains to be rod-like as observed by SEM images (Supplementary Fig. 14). The specific state of Cu species in Cu-DBC after electrocatalysis has been characterized. As shown in Supplementary Fig. 15a, the Raman spectrum of the fresh Cu-DBC is similar to that of the tested one with three main peaks centered around 234.4, 1346, and $1596 \, cm^{-1}$ assigned to the Cu–$O_4$ site, D and G peaks for the graphite-like carbon skeleton, respectively. And there are no other new Cu oxides, such as CuO or $Cu_2O$, signals appear[43,44], indicating that Cu-DBC did not transform into Cu oxides during electrocatalysis. In addition, the Raman spectrum was simulated by density functional theory (DFT) calculations (Supplementary Fig. 15b). And the related stretching vibrations of the three main peaks are shown as Supplementary Movies 1–3. Moreover, the state of Cu species after electrocatalysis was also revealed by XPS (Supplementary Fig. 16). The Cu $2p$ and Cu LMM spectra of Cu-DBC electrodes after the test are similar to those of the fresh ones, revealing that the Cu species show no obvious change after ECR. In addition, no metallic Cu was generated after the electrocatalysis as there was hardly a signal appearing at ~567 eV assigned to Cu(0) from the Auger Cu LMM region shown in Supplementary Fig. 16b. And the broad region from 569 to 571 eV belongs to Cu(II) and Cu(I) species[12]. As shown in Supplementary Fig. 16a and Supplementary Table 4, the state of Cu in Cu-DBC and the ratio of Cu(II) to Cu(I) almost remains to be unchanged after the electrocatalysis.

**ECR performances of crystalline single-site Cu electrocatalysts.** To explore the relationship between the coordination environment of the metal site and the selectivity of electrocatalysis over the single-site Cu electrocatalysts, the ECR performances of several other crystalline framework materials that also contain Cu single sites with electron-donating ligands were further investigated. The conductive MOF of Cu-HHTP with Cu–$O_4$ sites, and another two COFs of Cu-TTCOF and Cu-PPCOF that contain the most widely investigated Cu–$N_4$ sites of metalloporphyrin

and metallophthalocyanine, respectively, have been synthesized and utilized for ECR test. As shown in Supplementary Figs. 17–19, these three electrocatalysts synthesized by the reported methods exhibit similar XRD patterns to the simulated ones[45–47]. The structures of tested crystalline electrocatalysts with single Cu sites are shown in Fig. 3a. For the initial performance comparison, LSV curves were firstly recorded to assess the electrocatalysis activity under the applied potential varying from 0 to $-1.0 \, V$ vs. RHE in 1 M KOH electrolyte. According to the polarization curves shown in Fig. 3b, Cu-DBC and Cu-HHTP show higher current densities than those of Cu-TTCOF and Cu-PPCOF over the entire potential range. Cu-DBC shows the largest total current densities during the test among all and presents a high value of $-348 \, mA \, cm^{-2}$ at $-1.0 \, V$ vs. RHE, indicating superior activity towards electrocatalysis. The ECR performances of Cu-HHTP, Cu-TTCOF, and Cu-PPCOF were evaluated under the same conditions as for Cu-DBC. As shown in Supplementary Fig. 20b, the $FE_{ECR}$ of the Cu-HHTP during ECR test is over 65.0%, indicating inhibited HER over the entire potential range. However, the ECR selectivity of Cu-HHTP toward a single product is not satisfied with FEs of $CH_4$ and $C_2H_4$ reached ~42.6% and ~40.9% at $-0.9 \, V$ vs. RHE, respectively. Despite the low selectivity of Cu-HHTP to $CH_4$, it still generates hydrocarbons with a highest FE of ~83.5% at $-0.9 \, V$ vs. RHE (Supplementary Fig. 20). Furthermore, the electrochemical impedance spectroscopy (EIS) tests were recorded under $-0.9 \, V$ vs. RHE to reveal the electrocatalytic kinetics on the electrode/electrolyte surface of Cu-DBC and Cu-HHTP during ECR. As shown in the Nyquist plots of Supplementary Fig. 21, Cu-DBC exhibits smaller charge transfer resistance ($R_{ct}$) of $1.73 \, \Omega$ than that of $6.89 \, \Omega$ for Cu-HHTP, implying faster electron transfer and enhanced activity for ECR on Cu-DBC.

The FEs of ECR on the Cu-TTCOF and Cu-PPCOF are all less than 55.0%, as shown in Supplementary Figs. 22 and 23. The products generated from the electrocatalysis over Cu-PPCOF are dominant $H_2$ with a small amount of $CH_4$ and CO (Supplementary Fig. 22). The main ECR product of Cu-DBC, Cu-HHTP, Cu-TTCOF, and Cu-PPCOF is all $CH_4$, and the FEs of $CH_4$ at $-0.9 \, V$ vs. RHE are 80%, 42.6%, 43.6%, and 6.34% for the four crystalline electrocatalysts, respectively (Fig. 3c). The partial current densities of $CH_4$ at different potentials are calculated to reveal the excellent catalytic activity of ECR-to-$CH_4$ over Cu-DBC. The Cu-DBC displays a high partial $CH_4$ current density of $-162.3 \, mA \, cm^{-2}$ at $-0.9 \, V$ vs. RHE, much higher than those of the other three single-site Cu electrocatalysts (Fig. 3d). Additionally, the enhanced ECR performance may attribute to electrochemically active surface area (ECSA). The ECSA of catalysts is related to the effective specific surface area in the test system, which can identify by the double layer capacitance ($C_{dl}$) obtained from cyclic voltammetry curves. As calculated from Supplementary Figs. 24–27, $C_{dl}$ of the Cu-DBC, Cu-HHTP, Cu-TTCOF, and Cu-PPCOF are 7.72, 6.28, 5.05, and $1.06 \, mF \, cm^{-2}$, respectively. It indicated that the Cu-DBC catalyst contains much accessible catalytic area and more effective active sites for electrocatalysis, leading to enlarged ECR performance. The overall ECR-to-$CH_4$ performances of the four single-site Cu electrocatalysts under $-0.9 \, V$ vs. RHE have been concluded as a radar chart shown in Fig. 3e in which the Cu-DBC presents the largest shape area with the best performance in the four crystalline porous catalysts.

The XRD patterns and morphologies of these three single-site Cu electrocatalysts before and after electrocatalysis also have been characterized and analyzed. As shown in Supplementary Fig. 28, there is no phase transition or obvious structural change of Cu-TTCOF and Cu-PPCOF after electrocatalytic tests. However, the structural transition occurs in 2D conductive Cu-HHTP where

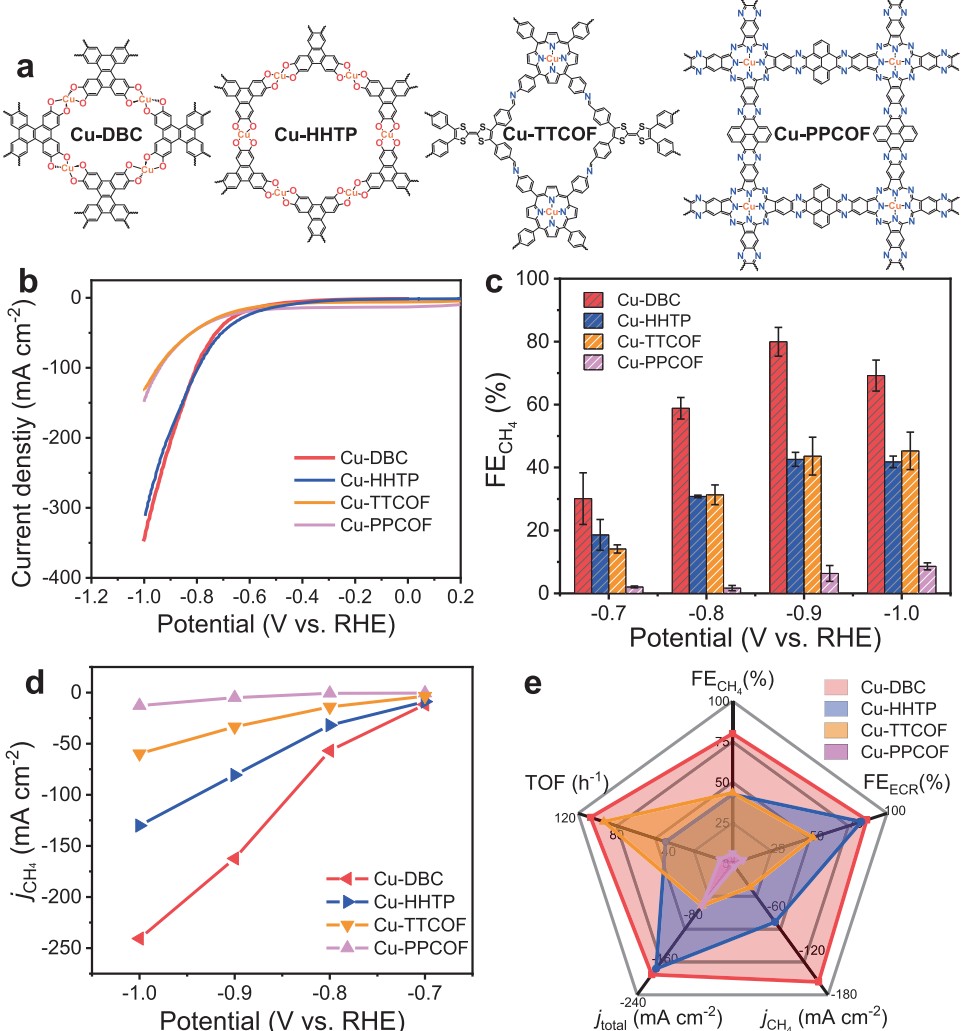

**Fig. 3 ECR performances of crystalline single-site Cu electrocatalysts. a** The formula diagrams. **b** LSV polarization curves. **c** Faradaic efficiencies (FEs) for $CH_4$ recorded at different applied potentials. **d** Partial $CH_4$ current densities. **e** Overall ECR performance evaluation at $-0.9$ V vs. RHE.

$Cu_2O$ phase generates after the electrolysis under reductive potential as shown in Supplementary Fig. 28a, which is consistent with the previous research[42]. Additionally, the morphologies of the three single-site Cu electrocatalysts were revealed by SEM. The Cu-TTCOF and Cu-PPCOF exhibit almost no obvious morphological change after electrocatalysis. The appearance of Cu-HHTP, however, tends to collapse after the test (Supplementary Fig. 29b). The structural change and the generation of $Cu_2O$ may lead to the poor selectivity of Cu-HHTP catalyst towards ECR to a single product.

**Discussion**

The catalytic mechanism of Cu-DBC were further investigated. According to the characterizations of the fresh and tested Cu-DBC catalysts, there is no Cu(0) or Cu oxides generated during the ECR. The high value-added products usually generate over the metallic Cu or Cu(I) species according to the previous reports[12,28,38,42]. Except for superior conductivity to the conventional MOF, the cMOF usually possesses unique redox properties owing to the highly conjugated ligands that could reversibly loss or gain electrons accompanied by the valence change of the coordination metal atoms. It has been reported in other electrocatalysis reactions that the generated intermediate redox states of the catalyst might provide active catalytic sites[48,49].

Therefore, the redox properties of Cu-DBC have been investigated. As shown in Supplementary Fig. 30, the cyclic voltammogram (CV) curves of Cu-DBC were recorded with the voltage window containing ECR reaction region or redox region. The redox peaks of both CV curves are almost the same, suggesting similar reversible redox properties of Cu-DBC with or without ECR process. The CV curve tested with the voltage window containing both ECR reaction and redox regions is analyzed as shown in Supplementary Fig. 31. As displayed in Supplementary Fig. 31b, three couple of reversible redox peaks in redox region are assigned to different redox states shown in Supplementary Fig. 31c. Thus, it could be deduced that Cu(I) redox state of Cu-DBC is probably the active sites for the production of the eight-electron products, $CH_4$, during the ECR reaction.

To further investigate the catalytic mechanism, DFT calculations have been conducted to reveal the key aspect of the electrocatalytic selectivity over the single-site Cu electrocatalysts. As shown in Fig. 4b, it could be observed that the three electrocatalysts with different coordination environments of Cu atom, including Cu–$O_4$ site in Cu-DBC, porphyrin Cu–$N_4$ site in Cu-TTCOF, and phthalocyanine Cu–$N_4$ site in Cu-PPCOF, show different activity and selectivity towards ECR and HER. The results of CV tests and the reported conclusions suggest that reductive Cu species with relatively low valence tend to facilitate

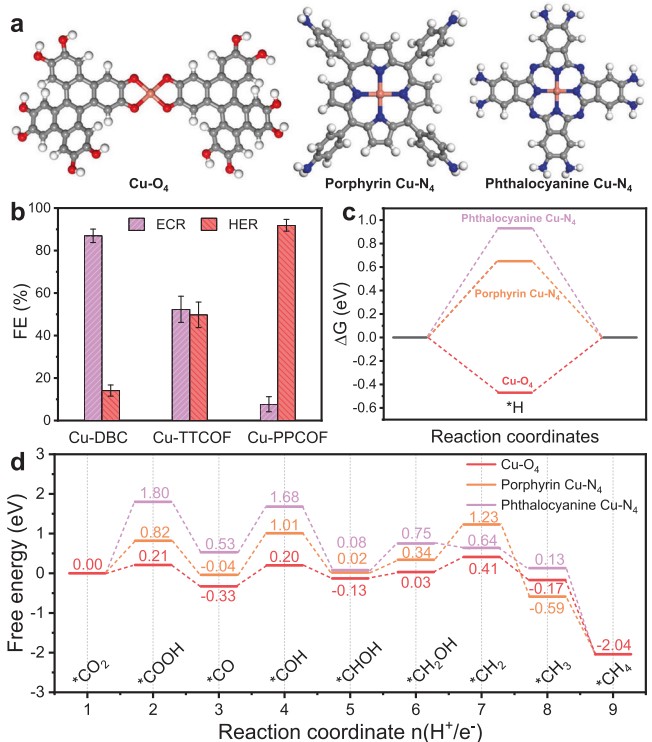

**Fig. 4 The dependence of ECR selectivity on Cu-site coordination environment. a** Schematic of DFT calculation models for Cu–O₄, porphyrin Cu–N₄, and phthalocyanine Cu–N₄ sites in the Cu-DBC, Cu-TTCOF, and Cu-PPCOF catalysts. Gray, blue, red, white, and orange spheres represent C, N, O, H, and Cu atoms, respectively. **b** FEs for ECR and HER recorded at −0.9 V vs. RHE on different single Cu site electrocatalysts. **c** The energy profiles of *H adsorption on different catalytic sites by DFT calculations. **d** Free energy profiles for the ECR-to-CH₄ reaction pathway.

$CO_2$ reduction to generate high value-added products. Therefore, the Cu sites should accept ($H^+/e^-$) pairs beforehand to achieve higher electron densities of Cu species for the following ECR process. The energy profiles of different Cu sites for receiving a ($H^+/e^-$) pair have been given in Fig. 4c. It was found that the *H adsorption site is either the oxygen or nitrogen atoms rather than Cu atoms in the Cu sites (Supplementary Fig. 32). And the adsorption of *H on the O atom in Cu–O₄ site is much preferred in energy while it is thermodynamically unfavorable on N atoms in both porphyrin Cu–N₄ or phthalocyanine Cu–N₄. It suggests that the Cu species in Cu–N₄ sites are not easy to be reduced compared with those in Cu–O₄ sites, and thus Cu-TTCOF and Cu-PPCOF show poorer ECR selectivity in contrast with Cu-DBC. In addition, the charge analysis also reveal that the Cu atom shows higher electron density after the Cu–O₄ site accept ($H^+/e^-$) pairs, which is beneficial to the $CO_2$ adsorption and reduction (Supplementary Table 5). Meanwhile, the formation of H₂ (i.e. full HER process) is restricted for Cu–O₄ system due to the high-energy barrier of 1.27 eV (Supplementary Fig. 33) following the Volmer–Tafel mechanism, indicating that the catalytic structure can be Cu–O₄Hₓ ($x = 1$–4) under reaction conditions. The activation of phthalocyanine Cu–N₄ is the most thermodynamically unfavorable among all (Fig. 4c). The energy profiles presented in Supplementary Fig. 34 imply that during the activation process of the Cu-PPCOF catalyst, the *H tends to interact with the linkage N atoms rather than N atoms in Cu–N₄ sites, indicating that the electroreduction active sites in Cu-PPCOF is the linkage N atoms. As the doped N atoms in the carbon skeleton tend to generate hydrogen, the Cu-PPCOF shows much higher HER selectivity than that of ECR.

After revealing the difference in selectivity, DFT calculations have been subsequently conducted to reveal the detailed free energy profiles for each reaction coordinate during ECR-to-CH₄ pathways on different single Cu sites. It is generally accepted that there are two possible reaction pathways for $CO_2$ reduction to CH₄, the *HCOOH pathway or the *CO pathway. As the main ECR products for the three electrocatalysts are all CH₄ and CO, and the CO production decreases with the formation of CH₄, it could be deduced that an eight-electron pathway involved with *CO intermediate for ECR-to-CH₄ occurs on these investigated Cu catalysts. The Gibbs free energy diagrams and structures for each dominant intermediate are summarized in Fig. 4d and Supplementary Figs. 35 and 36. The specific ECR-to-CH₄ reaction pathway, *CO₂ → *COOH → *CO → *COH → *CHOH → *CH₂OH → *CH₂ → *CH₃ → *CH₄ (* represents the catalytic site), involves the reduction of $CO_2$ along with ($H^+/e^-$) pairs and the desorption of H₂O. As displayed in Fig. 4d, the tendencies of free energy changes on Cu–O₄ site (the red lines) and porphyrin Cu–N₄ site (the orange lines) are analogous. And during the whole reaction pathway, four reaction steps are not spontaneous ($\Delta G > 0$), including *CO₂ → *COOH, *CO → *COH, *CHOH → *CH₂OH, and *CH₂OH → *CH₂. The step that a ($H^+/e^-$) attacks the carbon atom of *CO to form *COH intermediate exhibiting the largest free energy increase is the potential determining step (PDS). The free energy barriers of Cu-DBC and Cu-TTCOF for PDS are 0.53 and 1.05 eV, respectively. In addition, the Cu-DBC electrocatalyst with Cu–O₄ sites also possesses the lowest energy barriers of the other three energetically unfavorable steps, implying the better ECR-to-CH₄ reaction selectivity and activity. As it comes to the phthalocyanine Cu–N₄ site in Cu-PPCOF, the PDS is the first step (*CO₂ → *COOH) with an energy barrier as high as 1.80 eV, implying the unfavorable ECR process. Moreover, the energy barrier of *CO → *COH step on phthalocyanine Cu–N₄ site (1.15 eV) is also much higher than that on Cu–O₄ (0.53 eV). Additionally, all the free energy profiles in Fig. 4d can be drawn in the way with applied potentials ($U_{Onset} = -\Delta G_{max}/e$) involved. As shown in Supplementary Figs. 37–39, all three materials are active for the reaction with downhill schemes under $U_{Onset}$. It can also be noticed that the Cu–O₄ system shows the smallest applied potential of −0.53 V and the other two systems show much higher onset potentials for $CO_2$ reduction. Therefore, in contrast with the other two nitrogen coordination single Cu site catalysts (Cu-TTCOF and Cu-PPCOF), the particularity of Cu–O₄ site in Cu-DBC is that it is easier to be reduced into low-valence Cu site during the activation process and more energetically favorable for the following $CO_2$ reduction.

In conclusion, this work provides an electrocatalyst for $CO_2$ reduction based on Cu-based cMOF. The synthesized Cu-DBC containing dispersed single Cu sites and uniform micropores shows redox properties and $CO_2$ adsorption capacity. Combined with GDL, the Cu-DBC catalyst exhibits high selectivity to methane of ~80% at −0.9 V vs. RHE as well as a large catalytic current density of ~−203 mA cm⁻² when tested in the flow cell, which is one of the state-of-art ECR-to-CH₄ catalysts to date. The correlation between ECR selectivity and single Cu site coordination environment was investigated via DFT calculations and electrocatalytic measurements. The Cu–O₄ sites in Cu-DBC with low reaction energy barrier show better ECR performance compared to those nitrogen-coordinated Cu sites. The design of proposed conductive MOF may establish theoretical and experimental foundations for a better understanding of the correlation between the catalyst structure and electrochemical performance of ECR for further research.

## Methods

**Synthesis of Cu-DBC**. The synthesis of Cu-DBC was reported[34]. 8OH-DBC (8.6 mg) and of Cu(OAc)$_2$·H$_2$O (6.0 mg) were dispersed in the mixture of degassed dimethyformamide and deionized water. This vial was placed in 85 °C oven for 72 h. The reactant was washed with water and acetone several times, and dried overnight in vacuum at room temperature to obtain the black product.

**Synthesis of Cu-HHTP**. The synthesis of Cu-HHTP was reported[45]. A solid mixture of HHTP (7.5 mg) and Cu(C$_5$H$_4$F$_3$O$_2$)$_2$ (10.5 mg) was dissolved in 1 mL of deionized water. Then, 0.10 mL of NMP was added dropwise. The reaction mixture was heated in an isothermal oven at 85 °C for 12 h resulting in dark blue crystals. The obtained crystals were washed with deionized water, and then acetone and dried in air.

**Synthesis of Cu-TTCOF**. The synthesis of Cu-TTCOF followed our previously reported method[46]. The mixture of Cu-TAPP (14.8 mg), TTF-4CHO (12.4 mg), 1,4-dioxane (0.5 mL), and 1,3,5-trimethylbenzene (0.5 mL) was sonicated to dissolve, and then 6 M aqueous acetic acid (0.2 mL) was added. The mixture was heated at 120 °C and left undisturbed for 72 h. The obtained sample was transferred to a Soxhlet extractor and washed with THF and acetone. Finally, the product was evacuated at 150 °C under dynamic vacuum overnight.

**Synthesis of Cu-PPCOF**. The synthesis of Cu-PPCOF was conducted via a modified method according to the previous reported[47]. [NH$_2$]$_8$CuPc (17.5 mg) and PDQ (15.0 mg) were added in the mixture of dimethylacetamide and ethylene glycol with acetic acid catalyst at 200 °C. After 1 week, Cu-PPCOF was collected as gray-green powder. The obtained sample was transferred to a Soxhlet extractor and washed with THF and acetone. Finally, the product was evacuated at 80 °C under dynamic vacuum overnight.

**Characterizations and instruments**. PXRD patterns were recorded on a D/max 2500VL/PC diffractometer (Japan) equipped with a graphite monochromatized Cu Kα radiation source (λ = 1.54060 Å). The corresponding working voltage and current are 40 kV and 150 mA, respectively. TEM and HR-TEM images were recorded on a JEOL-2100F apparatus at an accelerating voltage of 200 kV. Morphological and microstructural analyses were conducted using a SEM (JSM-7600F) at an accelerating voltage of 10 kV. EDS was performed with a JSM-5160LV-Vantage type energy spectrometer. Nitrogen adsorption–desorption isotherms were recorded at 77 K using a Micromeritics instrument (Micromeritics ASAP2020 analyzer). The pore size distribution was calculated by nonlocal density functional theory (NLDFT). CO$_2$ adsorption isotherms were degassed in vacuum at 120 °C for 12 h and then measured by Quantachrome Instruments Autosorb IQ2 at 298 K. Thermogravimetric analysis of Cu-DBCs powder samples was performed on a Diamond TG/DTA/DSC Thermal Analyzer System (Perkin-Elmer, USA) with a heating rate of 10 °C min$^{-1}$ to 800 °C under air atmosphere. XPS was performed on a scanning X-ray microprobe (PHI 5000 Versaa, ULAC-PHI, Inc.) using Al Kα radiation and the C 1s peak at 284.8 eV as the internal standard. ICP-AES (Prodigy, USA) was used to measure the content of metal ions. The $I$–$V$ profiles tests were conducted with a probe station at room temperature (25 °C) under ambient conditions with a computer-controlled analog-to-digital converter (2636B, Kethley). The conductive sample was pressed into a sheet using a ton of pressure in the mold and the test voltage was among the range from −5.0 to 5.0 V. Nuclear magnetic resonance (NMR) was carried out on an AVANCE III HD-400. The gases (CO, CH$_4$, C$_2$H$_4$, and H$_2$) were detected and analyzed by gas chromatography (7820A, Aglient) equipped with an FID and a TCD. The isotope-labeled experiments were performed using $^{13}$CO$_2$, and the products were analyzed using gas chromatography-mass spectrometry (7890B and 5977B, Aglient). The electrochemical measurements were carried out using an electrochemical workstation (Bio-Logic).

**Electrochemical measurements**. Ten milligrams of electrocatalyst was grinded to powered and then dispersed into 950 μL H$_2$O/ethanol mixture with 50 μL Nafion. Fifty microliters of the catalyst ink was dropped onto a commercial GDL electrode to form the work electrode. CO$_2$ electroreduction was performed in a three-electrode flow cell using 1 M KOH aqueous electrolyte. The Ag/AgCl electrode (saturated KCl) was used as the reference electrode, and the counter electrode was matched with a Pt plate. An anion exchange membrane was used to separate the cathode and anode.

## Data availability

The data that support the findings of this study are available within the paper and its supplementary information files or are available from the corresponding authors upon reasonable request. Source data are provided with this paper.

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

## Acknowledgements

This work was financially supported by NSFC (No. 21871141, 21871142, 21701085, and 21901122); the NSF of Jiangsu Province of China (No. BK20171032); the Natural Science Research of Jiangsu Higher Education Institutions of China (No. 17KJB150025 and 19KJB150011); and Project funded by China Postdoctoral Science Foundation (No. 2018M630572 and 2019M651873); Priority Academic Program Development of Jiangsu Higher Education Institutions and the Foundation of Jiangsu Collaborative Innovation Center of Biomedical Functional Materials.

## Author contributions

Y.-Q.L., Y.Z. and L.-Z.D. conceived and designed the idea, analyzed the data, and discussed the result and prepared the manuscript. L.-Z.D., Y.Z., X.H., J.N.C. and J.Z. synthesized the electrocatalysts. Y.Z. and S.L. conducted the characterizations and designed the electrocatalytic $CO_2$RR-related experiments. J.-H.W. assisted with the characterizations. Y.Z. wrote the manuscript. All the authors reviewed and contributed to this paper.

## Competing interests

The authors declare no competing interests.
