## [Peer Review File · Nature Communications]

Coordination Environment Dependent Selectivity of
Single-Site-Cu Enriched Crystalline Porous Catalysts in CO₂
Reduction to CH₄REVIEWER COMMENTS

Reviewer #1 (Remarks to the Author):

This work reports on the study of a CuO₄-based MOF exhibiting catalytic properties for CO₂ reduction to CH₄ superior compared with previously reported Cu-based MOFs.

The work is well performed with a full experimental characterization using many complementary techniques and the results are convincing. I recommend publication of this manuscript in Nature Communication once the english has been polished. I have found several typos and many unclear sentences.

More minor comments:

1. valent change  valence change

2. please correct Faradic efficiency throughout the text

3. It is claimed that the samples show a good cristallinity but the patterns of Figs S16-S17 show a poor cristallinity

4. Please better explain/rephrase the following sentences:

1... almost no Cu-DBC transforms into Cu oxides during electrocatalysis ...

2... Thus, it could be deduced that during the ECR reaction, Cu(I) redox state of Cu-DBC may attribute to the generation of multi-electron transfer products of CH₄.

3... As the main ECR products for the three electrocatalysts are all CH₄ and CO, and the CO production decreases with the formation of CH₄. It could

be deduced that an eight-electron pathway involved with *CO intermediate for ECR-to-CH₄ occurs on these investigated Cu catalysts.  is this one sentence?

4. Please split this sentence into two and make them more readable.

As displayed in Figure 4d, The tendencies of free energy changes on Cu-O₄ site (the red lines) and porphyrin Cu-N₄ site (the orange lines) are analogous, and in the whole reaction pathway, four reaction steps are not spontaneous with $\Delta G > 0$, including *CO₂ → *COOH, *CO → *COH, *CHOH → *CH₂OH, and *CH₂OH → *CH₂, in which the step with the largest free energy increase that a (H⁺/e⁻) attacks the carbon atom of *CO to form *COH intermediate is the potential determining step (PDS).

5. On the contrary, the 2D conductive Cu-HHTP occurs structure transition and Cu₂O phase generated after the electrolysis ...  please clarify the sentence - review the English

5. It should be clarified which kind of models are used for the DFT calculations (periodic? fragments?)  e.g. the results shown in Fig. 4. If fragment models are used then a lot of missing info would be required.

6. Table S3 has a type of the Lowdin charge.

Reviewer #2 (Remarks to the Author):

In the submitted manuscript, the authors have reported a Cu-based conductive MOF (Cu-DBC) that shows promising activity for the selective reduction of CO₂ to CH₄ when incorporated into a gas-diffusion electrode and assembled into a flow cell. The authors demonstrate high activity and selectivity for CH₄ production at -0.9 V vs RHE, with 80% FE for CH₄ and a partial current density of ~160 mA cm⁻². The authors conducted ¹³C-labeled experiments to confirm that all CH₄ is coming from the ¹³C-labeled CO₂, and they conducted postmortem characterization of the Cu-DBC showing the MOF maintains its crystallinity throughout the electrolysis. They also show the system has stable performance for CH₄ production up to ~2.5 h. The authors do direct comparison with other Cu-based MOF systems, and conclude with a series of DFT studies to propose a mechanism for the CO₂RR on the Cu-DBC system.

Overall, this is a well-considered and well-written manuscript, and I believe it has sufficiently novelty and impact for publication in Nature Communications. The material is well characterized, the authors performed the crucial postmortem characterization to ensure the integrity of the synthesized unit during the electrochemical measurements, the $^{13}\text{CO}_2$ labeling experiments confirm the CH_4 product comes from CO_2 reduction, and they compared the system to known catalysts. Although CH_4 is not a commercially viable desired product, it is nevertheless a more highly reduced and scientifically meaningful product than CO (8 e⁻ instead of 2 e⁻). The DFT calculations may provide some insight into the mechanism of the CO_2 reduction on this material, although such calculations are not sufficient to assign unambiguously the binding states and reaction pathways without supporting in situ X-ray absorbance spectroscopy or equivalent spectroscopic measurements. I believe this manuscript is appropriate for publication based on the merits of the experimental studies. I saw no major issues that would prevent publication, although I would suggest the authors do one more proofread for grammatical inconsistencies.

Reviewer #3 (Remarks to the Author):

This manuscript reports a metal-organic framework electrocatalyst for CO_2 to methane production. This cathode shows a high Faradaic efficiency of ca. 80 % at a high overpotential region. The contents could be interesting however the manuscript does not sufficiently provide data enough to support the performance of their electrode materials. Therefore, this referee cannot recommend editor to publish this manuscript in this journal. The detailed reasons are described below.

1. Characterizations are not well discussed and data is not well considered. For example, a conductivity of $1.2 \times 10^{-2} \text{ S m}^{-1}$ is not high rather low. From point of the selection of linker, an electrically-conductive Cu-DBC would be unlikely because electrons will be trapped at oxygen and this leads to a semiconductor. Furthermore, the sentence “which is one of the highest conductivities of cMOFs” at the line 137 is wrong. See for example “<https://doi.org/10.1021/acs.chemrev.9b00766>”. In addition, the Raman spectra in Figure S13 is quite similar to that of carbon materials. Is this data taken with carbon or without carbon? This referee notes that a calculation of theoretical Raman spectra is quite easy if the authors can perform DFT.

2. -0.9 V vs RHE is a quite reductive atmosphere and usually this condition can be used to clean electrolyte to decompose organic contaminations. Therefore, it is quite surprise that Cu-DBC survives in this condition. The authors should carefully check the electrode after the electrochemical test.

3. The DFT calculation (Figure 4) and the related discussions do not support the results. For example, Figure 4c cannot explain the selectivity and activity because of no application of potential. Because of same reason, Figure 4d explains nothing about the electrochemical properties of Cu-DBC but shows that all three materials are inactive for the reaction because of uphill schemes.

Point-by-point response to the reviewers' comments

Reviewer #1:

This work reports on the study of a CuO₄-based MOF exhibiting catalytic properties for CO₂ reduction to CH₄ superior compared with previously reported Cu-based MOFs.

The work is well performed with a full experimental characterization using many complementary techniques and the results are convincing. I recommend publication of this manuscript in *Nature Communication* once the English has been polished. I have found several typos and many unclear sentences.

Reply: Thanks for the valuable comments. We thank the reviewer has recommended the publication of our research work in *Nature Communications*. The grammar and writing were checked and corrected throughout the manuscript. In addition, we further improved the manuscript with supplementary experiments and theoretical calculations according to the suggestions of the reviewers. The revised parts were highlighted in yellow in the submitted revised version.

More minor comments:

1. valent change  valence change

Reply: Thank you for the valuable suggestion. The “valent change” was corrected into “valence change” throughout the whole manuscript.

2. please correct Faradaic efficiency throughout the text.

Reply: Thanks for the valuable suggestion. The right spelling of “Faradaic efficiency” was used in the revised manuscript.

3. It is claimed that the samples show a good crystallinity but the patterns of Figs S16-S17 show a poor crystallinity.

Reply: Thanks for the insightful comment. The two samples were successfully synthesized according to the previously reported methods, and the XRD patterns are consist with the simulated ones. We remeasured the XRD patterns with slower step size using monocrystalline silicon as substrates to reduce the noise of the signal. **Supplementary Fig. 16~17** were replaced as below. And the related description sentence was revised as “these three electrocatalysts synthesized by the reported methods exhibit similar XRD patterns to the simulated ones” in the manuscript.

Supplementary Fig. 16 Characterizations of the as-prepared Cu-TTCOF. **a** XRD patterns and **b** SEM image.

Supplementary Fig. 17 Characterizations of the as-prepared Cu-PPCOF. **a** XRD patterns and **b** SEM image.

4. Please better explain/rephrase the following sentences:

Reply: Thanks for the valuable comments and suggestions. The meanings of the following sentences were explained and they were rephrased for clearly reading.

(1) The sentence of "... almost no Cu-DBC transforms into Cu oxides during electrocatalysis ..." means that "according to the Raman spectra of the fresh and tested Cu-DBC, it could be seen that there is no Cu oxide generating during electrocatalysis". It was changed into "...Cu-DBC did not transform into Cu oxides during electrocatalysis..." in the revised manuscript.

(2) The sentence of "... Thus, it could be deduced that during the ECR reaction, Cu(I) redox state of Cu-DBC may attribute to the generation of multi-electron transfer products of CH₄" means that the Cu(I) redox state of Cu-DBC is probably the active catalytic sites and attribute to the generation of CH₄ (the eight-electron products of ECR). As the Cu(I) is the active sites for multi-electron products, and Cu(II) species transfer into Cu(I) species due to the excellent redox properties in the Cu-DBC system, it could be deduced that the generated Cu(I) redox state is the active sites for ECR

to produce multi-electron products. Therefore, this sentence has been rephrased into “Thus, it could be deduced that Cu(I) redox state of Cu-DBC is probably the active sites for the production of the eight-electron products, CH₄, during the ECR reaction.”

(3) ... As the main ECR products for the three electrocatalysts are all CH₄ and CO, and the CO production decreases with the formation of CH₄. It could be deduced that an eight-electron pathway involved with *CO intermediate for ECR-to-CH₄ occurs on these investigated Cu catalysts.  is this one sentence?

Reply: Thanks for the valuable question. This is indeed one sentence, and it has been corrected into “As the main ECR products for the three electrocatalysts are all CH₄ and CO, and the CO production decreases with the formation of CH₄, it could be deduced that an eight-electron pathway involved with *CO intermediate for ECR-to-CH₄ occurs on these investigated Cu catalysts.” in the revised manuscript.

(4) Please split this sentence into two and make them more readable.

As displayed in Fig. 4d, The tendencies of free energy changes on Cu-O₄ site (the red lines) and porphyrin Cu-N₄ site (the orange lines) are analogous, and in the whole reaction pathway, four reaction steps are not spontaneous with $\Delta G > 0$, including *CO₂ → *COOH, *CO → *COH, *CHOH → *CH₂OH, and *CH₂OH → *CH₂, in which the step with the largest free energy increase that a (H⁺/e⁻) attacks the carbon atom of *CO to form *COH intermediate is the potential determining step (PDS).

Reply: Thanks for the valuable suggestion. The sentence has been divided into “As displayed in Fig. 4d, the tendencies of free energy changes on Cu-O₄ site (the red lines) and porphyrin Cu-N₄ site (the orange lines) are analogous. And during the whole reaction pathway, four reaction steps are not spontaneous ($\Delta G > 0$), including *CO₂ → *COOH, *CO → *COH, *CHOH → *CH₂OH, and *CH₂OH → *CH₂. The step that a (H⁺/e⁻) attacks the carbon atom of *CO to form *COH intermediate exhibiting the largest free energy increase is the potential determining step (PDS)” in the revised manuscript.

5. On the contrary, the 2D conductive Cu-HHTP occurs structure transition and Cu₂O phase generated after the electrolysis ...  please clarify the sentence - review the English

Reply: Thanks for the valuable suggestion. It has been rephrased as “However, the structural transition occurs in 2D conductive Cu-HHTP where Cu₂O phase generates after the electrolysis”

6. It should be clarified which kind of models are used for the DFT calculations (periodic? fragments?)  e.g. the results shown in Fig. 4. If fragment models are used then a lot of missing info would be required.

Reply: Thanks for the insightful suggestion. The DFT calculations are based on fragment models which are extracted from crystal structures of the three systems and are reoptimized. We checked the DFT calculation details and the fragment model of Cu-O₄ in Cu-DBC was refined to be accurate in the revised version as shown in **Fig. R1**. The related data in **Fig. 4a**, **Supplementary Fig. 30**, **Supplementary Fig. 31**, and **Supplementary Fig. 33** were also updated based on the latest model in **Fig. R1**.

The structures of fragment models of the three systems were shown in **Fig. 4a** and the related coordinates were added in the **Computational methods** part in the revised supporting information. Periodic calculations are not performed due to the large unit cells, containing thousands of atoms, for example, 1408 atoms for Cu-DBC system (**Fig. R2**). However, the catalytic property can be localized on the metal centers and the long-rang interaction can be neglected for catalytic reactions. Thus, the selected fragments that contain the metal active sites are sufficient for the discussion in this work.

Fig. R1. The updated structure fragment model of Cu-O₄ site in Cu-DBC system.

Fig. R2. Schematic picture of the unit cell of Cu-DBC system, containing 1408 atoms.

In the revised supporting information, the detail computational methods, structure models (with coordinates) as well as the structures of intermediates (**Supplementary Fig. 33 and 34**) were given. And the related descriptions in **Computational methods** were rewritten as “In the present work, all the calculations including structure optimization, free energy calculation, and Raman properties were performed by using DFT (density functional theory) method with fragment models. The structure models (with coordinates) of the three systems are shown as below. The calculations were performed using the ORCA package employing the resolution of identity approximation.⁵ All the DFT calculations were performed using the Becke’s three-parameter hybrid functional with gradient corrections provided by Lee, Yang, and Parr (B3LYP) functional. Basis sets of def2-SVP were used to optimize the structures and def2-TZVP^{6,7} were adopted for Cu, C, N, O and H atoms in the complexes with decontracted auxiliary def2-TZVP/J Coulomb fitting basis sets to correct the energies.⁸ The DFT grid was set to GRID4, and the convergence threshold TIGHT was employed for the self-consistent field (SCF) and the optimization procedure. D3 dispersion correction developed by Grimme is included for weak interactions.⁹ Vibrational frequency calculations of optimized structures were performed at the same level of theory to ascertain the presence of a local minimum, confirming that there is no imaginary frequency observed for the considered systems, and was used to generate the zero-point energies (ZPE), the Gibbs free energies (G) and zero-point corrections”. And structure models (with coordinates) were added in the revised supporting information before the **References** part.

7. Table S3 has a type of the Löwdin charge.

Reply: Thank for the valuable suggestion. The charge analysis was done directly by ORCA calculations with Löwdin population outputs, which is, in general, not rotationally invariant unless an initial atom-centered basis of pure spherical harmonics is used or the atomic orbitals on the same atom are pre-orthogonalized [*Int. J. Quantum. Chem.*, 2006, 106, 2065]. The title of **Table S3** was changed as below in the revised supporting information.

Table S3 Löwdin charge analysis of the Cu-O₄ and Cu-N₄ systems, the results are obtained from Löwdin population analysis by ORCA calculations.

Model	Löwdin Charge
Cu-O ₄	0.39
Cu-O ₄ H ₁	0.28
Cu-O ₄ H ₄	0.04
Porphyrin Cu-N ₄	0.18

Porphyrin Cu-N ₄ H ₁	0.12
Phthalocyanine Cu-N ₄	0.19

Reviewer #2:

In the submitted manuscript, the authors have reported a Cu-based conductive MOF (Cu-DBC) that shows promising activity for the selective reduction of CO₂ to CH₄ when incorporated into a gas-diffusion electrode and assembled into a flow cell. The authors demonstrate high activity and selectivity for CH₄ production at -0.9 V vs RHE, with 80% FE for CH₄ and a partial current density of ~160 mA cm⁻². The authors conducted ¹³CO₂ labeled experiments to confirm that all CH₄ is coming from the ¹³CO₂, and they conducted postmortem characterization of the Cu-DBC showing the MOF maintains its crystallinity throughout the electrolysis. They also show the system has stable performance for CH₄ production up ~2.5 h. The authors do direct comparison with other Cu-based MOF systems, and conclude with a series of DFT studies to propose a mechanism for the CO₂RR on the Cu-DBC system.

Overall, this is a well-considered and well-written manuscript, and I believe it has sufficiently novelty and impact for publication in *Nature Communications*. The material is well characterized, the authors performed the crucial postmortem characterization to ensure the integrity of the synthesized unit during the electrochemical measurements, the ¹³CO₂ labeling experiments confirm the CH₄ product comes from CO₂ reduction, and they compared the system to known catalysts. Although CH₄ is not a commercially viable desired product, it is nevertheless a more highly reduced and scientifically meaningful product than CO (8 e⁻ instead of 2 e⁻). The DFT calculations may provide some insight into the mechanism of the CO₂ reduction on this material, although such calculations are not sufficient to assign unambiguously the binding states and reaction pathways without supporting in situ X-ray absorbance spectroscopy or equivalent spectroscopic measurements. I believe this manuscript is appropriate for publication based on the merits of the experimental studies. I saw no major issues that would prevent publication, although I would suggest the authors do one more proofread for grammatical inconsistencies.

Reply: Thanks for the valuable comments. We thank the reviewer has taken consideration for the publication of our research work in *Nature Communications*. The grammar and writing were checked and corrected throughout the manuscript. In addition, we further improved the manuscript with additional experiments and theoretical calculations according to the suggestions of the other reviewers. The revised parts were highlighted in yellow in the submitted revised version.

Reviewer #3:

This manuscript reports a metal-organic framework electrocatalyst for CO₂ to methane production. This cathode shows a high Faradaic efficiency of ca. 80 % at a high overpotential region. The contents could be interesting however the manuscript does not sufficiently provide data enough to support the performance of their electrode materials. Therefore, this referee cannot recommend editor to publish this manuscript in this journal. The detailed reasons are described below.

Reply: We thank the reviewer raised rigorous and valuable comments. The manuscript was improved according the reviewer's concerns. The description of conductivity was revised for more rigorous presentation, and the DFT calculations were supplemented to the discussion of Raman spectra. The structural stability of the Cu-DBC was investigated with several post-test characterizations including XRD patterns, XPS spectra, SEM images and Raman spectra. At last, the concerns on DFT calculations were explained and discussed, in particular, the energy profiles of the reaction pathways with the consideration of the applied onset potential were given, which is consistent with the conclusions of the whole research. All the revised parts were highlighted in yellow in the revised version.

1.Characterizations are not well discussed and data is not well considered. For example, a conductivity of $1.2 \times 10^{-2} \text{ S m}^{-1}$ is not high rather low. From point of the selection of linker, an electrically-conductive Cu-DBC would be unlikely because electrons will be trapped at oxygen and this leads to a semiconductor. Furthermore, the sentence "which is one of the highest conductivities of *c*MOFs" at the line 137 is wrong. See for example "<https://doi.org/10.1021/acs.chemrev.9b00766>” In addition, the Raman spectra in **Supplementary Fig. 13** is quite similar to that of carbon materials. Is this data taken with carbon or without carbon? This referee notes that a calculation of theoretical Raman spectra is quite easy if the authors can perform DFT.

Reply: Thanks for the constructive comments and meaningful question.

(1) We carefully read the references about the conductivities of the *c*MOFs and found that the tested conductivity of Cu-DBC in our work is indeed not really remarkable. Most of the *c*MOFs possess the conductivities approximately ranging from 10^{-6} to 10^4 S/m. In addition, the *c*MOFs with oxygen-functionalized conjugated linkers usually show lower conductivities than those contain nitrogen-functionalized or sulfur-functionalized conjugated linkers. However, since Cu oxides are accepted as well-performed CO₂ reduction electrocatalysts, and the reported single-site CO₂ catalysts are usually nitrogen-coordinated, in this research, we still selected oxygen-functionalized conjugated linker to synthesize the crystalline *c*MOFs with

oxygen-coordinated Cu sites and thus to investigate the difference of CO₂ reduction performance between the oxygen-coordinated and nitrogen-coordinated single Cu sites. Cu-DBC is a cMOF containing D₂-symmetric conjugated 8-OH-DBC ligand (Angew. Chem. Int. Ed. 2020, 59, 2-15, doi.org/10.1002/anie.202006102). The electrical conductivity of the synthesized Cu-DBC in this work is much higher than the conventional MOFs. It could afford the electron transfer during electrocatalysis, and thus achieve excellent CO₂ reduction activity. And it also showed good structural stability after electrocatalysis. Therefore, we selected Cu-DBC in this work. We are sorry for the inappropriate description that *Cu-DBC possesses one of the highest conductivities of cMOFs*, and the related sentence was corrected as “**Cu-DBC exhibits an electrical conductivity of $1.2 \times 10^{-2} \text{ S m}^{-1}$ which is much higher than those of the conventional MOFs without conjugated ligands.**^{34,36}” in the revised version.

(2) The Raman spectra in **Supplementary Fig. 13a** were obtained with the fresh and tested electrocatalysts without adding any other carbon materials. The Cu-DBC is a cMOF material formed by the combination of Cu nodes and conjugated DBC ligand, and the main components of its structure are conjugated carbon rings. The Raman test is more sensitive to carbon skeleton materials, and most of the materials with conjugated graphite-like structures show typical carbon signals, the D and G peaks. Thus, the Raman spectra of Cu-DBC exhibits the signal of D and G peaks.

In addition, according to the reviewer’s suggestion, we conducted the calculation of theoretical Raman spectrum. The unit cell of the system contains over a thousand atoms, thus a fragment model shown in **Fig. R1** that was extracted from crystal structures of the Cu-DBC system, is adopted instead periodic calculations. We applied harmonic approximation to obtain the vibrations of the system. The DFT simulations can provide a qualitative understanding of the Raman spectrum of Cu-DBC. As shown in the revised **Supplementary Fig. 13b**, the simulated Raman spectrum by DFT confirmed two main regions assigned to the Cu-O (pink region) and C fragment (blue region) stretching vibrations, respectively. The strong peaks in blue region around ~1380 (Peak 1) and ~1540 cm⁻¹ (Peak 2) are mainly attributed to the D and G peaks in C fragment as shown in **Supplementary Movie 1** and **Supplementary Movie 2**. The strongest signal of ~560 cm⁻¹ (Peak 3) is caused by the stretching vibrations of Cu-O shown in **Supplementary Movie 3**. Since the DFT calculation was conducted by using harmonic approximation based on the fragment rather than the whole structure, the intensity and the location of the peaks have a quantitative mismatch between theoretical calculations and experiments. However, the three categories of peaks (Cu-O, D and G) are consistent, and the locations of D and G peak are similar. Therefore, combining DFT calculations with experiments, we conclude that the Raman spectra that contain D and G peaks in this work are reasonable.

Fig. R1. The updated structure fragment model of Cu-O₄ site in Cu-DBC system.

Supplementary Fig. 13 Raman analysis. **a** Raman spectra of Cu-DBC before and after electrocatalysis. **b** Simulated Raman spectrum based on Cu-O₄ site fragment in Cu-DBC system.

The related descriptions of “The calculation of theoretical Raman spectrum of Cu-DBC was conducted by DFT calculations. The fragment model extracted from crystal structures of the Cu-DBC system is adopted. The harmonic approximation is applied to obtain the vibrations of the system. All these factors can cause a reasonable quantitative mismatch between theoretical calculations and experiments. The DFT simulations can provide a qualitative understanding of the Raman spectrum of Cu-DBC. As shown in the **Supplementary Fig. 13b**, the simulated Raman spectrum by DFT confirms two main regions assigned to the Cu-O (pink region) and C fragment (blue region) stretching vibrations, respectively. The strong peaks in blue region around ~1380 (Peak 1) and ~1540 cm⁻¹ (Peak 2) are mainly attributed to the D and G peaks in C fragment as shown in **Supplementary Movie 1** and **Supplementary Movie 2**. The strongest signal of ~560 cm⁻¹ (Peak 3) is caused by the stretching vibrations of Cu-O as shown in **Supplementary Movie 3**.” were added in the revised supporting information. **Supplementary Fig. 13** was changed as above, and **Supplementary Movie 1, 2 and 3** were supplemented in the revised version.

2. -0.9 V vs RHE is a quite reductive atmosphere and usually this condition can be used to clean electrolyte to decompose organic contaminations. Therefore, it is quite surprise that Cu-DBC survives in this condition. The authors should carefully check

the electrode after the electrochemical test.

Reply: Thanks for the insightful suggestion. The stability of the electrocatalyst under the test conditions indeed plays an important role in the catalytic mechanism analysis. Therefore, the structural stability of the electrocatalysts were taken into consideration to check whether the Cu-DBC could survive at the applied potential or not. We conducted a series of characterizations, including XRD patterns, XPS spectra, SEM images and Raman spectra, to evaluate the structure, morphology, and state of the Cu-DBC after electrocatalytic tests.

As shown in **Fig. 2f**, the XRD patterns of the fresh and tested electrode are consistent with each other, indicating that no phase transition or obvious structural change occurred in Cu-DBC after electrochemical tests. Furthermore, the morphologies of Cu-DBC still maintains to be rod-like as observed by SEM images shown in **Supplementary Fig. 12**. The Raman spectrum of the fresh Cu-DBC is similar to that of the tested one with, and there were no other new signals that belong to Cu oxides appear after electrochemical tests. Moreover, the state of Cu species before and after electrochemical tests was revealed by XPS analysis (**Supplementary Fig. 14** and **Supplementary Table 2**). The Cu 2p and Cu LMM spectra of Cu-DBC electrodes after the test are similar to those of the fresh ones, revealing that the Cu species show no obvious change after ECR. In addition, no metallic Cu was generated after the electrochemical tests as there was hardly a signal appearing at ~ 567 eV assigned to Cu(0) from the Auger Cu LMM region shown in **Supplementary Fig. 14b**. As shown in **Supplementary Fig. 14a** and **Supplementary Table 2**, the state of Cu in Cu-DBC and the ratio of Cu(II) to Cu(I) almost remains to be unchanged. All these post-test characterizations results suggest that the Cu-DBC could survive in the electrochemical conditions in this research.

Fig. 2f XRD patterns of fresh and tested Cu-DBC modified GDL-carbon paper electrodes.

Supplementary Fig. 12 SEM images of the Cu-DBC electrode, **a** before and **b** after electrocatalysis.

Supplementary Fig. 13a Raman spectra of Cu-DBC before and after electrocatalysis.

Supplementary Fig. 14 XPS analysis. **a** Cu 2p and **b** Auger Cu LMM XPS spectra of fresh and tested Cu-DBC modified GDL-CP electrodes.

Supplementary Table 2 Cu 2p XPS spectra peak fit parameters of Cu-DBC before and after electrocatalysis

Sample	Bind	Position	FWHM	Area (%)
Fresh	1-Cu(II)	933.77	1.92	84.6%
	2-Cu(I)	932.18	1.36	15.4%
After electrocatalysis	1-Cu(II)	933.77	1.92	86.9%
	2-Cu(I)	932.18	1.36	13.1%

3. The DFT calculation (Fig. 4) and the related discussions do not support the results. For example, Fig. 4c cannot explain the selectivity and activity because of no application of potential. Because of same reason, Fig. 4d explains nothing about the electrochemical properties of Cu-DBC but shows that all three materials are inactive for the reaction because of uphill schemes.

Reply: Thank you for the valuable comments. The computational hydrogen electrode (CHE) model that proposed by Nørskov et al. was applied to describe the Gibbs reaction free energy of reaction for CO₂RR elementary steps involving (H⁺ + e⁻) pair transfer [*Energy Environ. Sci.*, 2010, 3, 1311]. It can be noticed that the potential can be included by $\Delta G_n(U) = \Delta G_n(U=0) + neU$, where e is the elementary charge of an electron, n is the number of (H⁺ + e⁻) pairs transferred in CO₂RR and U is the electrode potential versus the reversible hydrogen electrode (RHE). It is generally accepted that the active Cu sites should gain a (H+e⁻) pair to reach a state with higher electron densities before CO₂ reduction. Therefore, the energy profiles of *H adsorption on different Cu catalytic sites have been calculated ahead of the CO₂ reduction as shown in **Fig. 4c**. It was intended to illustrate the free energy profile for $U = 0$, from which we can directly compare the free energy change of three systems. The results clearly show that the Cu-O₄ sites are more energetically favorable to reach an electron-rich state and easier to proceed CO₂ reduction, which is consistent with the electrochemical test results.

Since CO₂ reduction suffers from sluggish kinetics and the electrocatalysts are used to reduce rather than eliminate the energy barriers at current stage, it is reasonable that there are some uphill schemes. In **Fig. 4d**, the results show that the Cu-DBC electrocatalyst with Cu-O₄ sites possess the lowest energy barriers in the three systems for the non-spontaneous steps ($\Delta G > 0$), including *CO₂ → *COOH, *CO → *COH, *CHOH → *CH₂OH, and *CH₂OH → *CH₂. The agreement between theoretical calculations and experiments demonstrates that the selectivity and activity of the investigated three systems can be explained by the free energy changes based

on the CHE model.

In addition, according to the reviewer's comments, all the free energy profiles in **Fig. 4d** can be drawn in the way with applied potentials ($U_{\text{Onset}} = -\Delta G_{\text{max}}/e$) involved as shown in **Supplementary Fig. 35~37**, and all three materials are active for the reaction because of downhill schemes. It can also be noticed that the Cu-O₄ system shows the smallest applied potential of -0.53 V and the other two materials show much higher onset potentials for CO₂ reduction, which is also consistent with the conclusions of $U = 0$ V. Certainly, there are other more dedicated approaches to treat the applied potential, which can be very time-consuming for dealing with solid/electrolyte interfaces, especially with such large systems in this work. As the materials are tested with the same conditions, the current DFT models containing the main catalytic sites are reasonable to explain the catalytic mechanism, and the results of theoretical calculations and experiments are consistent, we think the current DFT calculations are convincing in this research.

Supplementary Fig. 35~37 were added in the revised supporting information. The related descriptions of “Additionally, all the free energy profiles in **Fig. 4d** can be drawn in the way with applied potentials ($U_{\text{Onset}} = -\Delta G_{\text{max}}/e$) involved. As shown in **Supplementary Fig. 35~37**, all three materials are active for the reaction with downhill schemes under U_{Onset} . It can also be noticed that the Cu-O₄ system shows the smallest applied potential of -0.53 V and the other two systems show much higher onset potentials for CO₂ reduction.” were added in the revised manuscript.

Supplementary Fig. 35 The energy profiles of Cu-O₄ sites in Cu-DBC with $U = 0$ and $U = U_{\text{Onset}}$.

Supplementary Fig. 36 The energy profiles of porphyrin Cu-N₄ in Cu-TTCOF with $U = 0$ and $U = U_{\text{Onset}}$.

Supplementary Fig. 37 The energy profiles of phthalocyanine Cu-N₄ in Cu-PPCOF with $U = 0$ and $U = U_{\text{Onset}}$.

REVIEWERS' COMMENTS

Reviewer #3 (Remarks to the Author):

This manuscript was remarkably improved. Therefore, this referee recommends to accept this work as it is.

A point-by-point response to the reviewer's comments

Reviewer #3:

This manuscript was remarkably improved. Therefore, this referee recommends to accept this work as it is.

Reply: Thanks for the valuable comments. We thank the reviewer has taken consideration for the publication of our research work in *Nature Communications*.